🔓 | **Open Peer Review** | Microbial Pathogenesis | Research Article

# *In vitro* lung epithelial cell model reveals novel roles for *Pseudomonas aeruginosa* siderophores

Donghoon Kang,[1] Qi Xu,[1,2] Natalia V. Kirienko[1]

**ABSTRACT**  The multidrug-resistant pathogen *Pseudomonas aeruginosa* is a common nosocomial respiratory pathogen that continues to threaten the lives of patients with mechanical ventilation in intensive care units and those with underlying comorbidities such as cystic fibrosis or chronic obstructive pulmonary disease. For over 20 years, studies have repeatedly demonstrated that the major siderophore pyoverdine is an important virulence factor for *P. aeruginosa* in invertebrate and mammalian hosts *in vivo*. Despite its physiological significance, an *in vitro*, mammalian cell culture model that can be used to characterize the impact and molecular mechanisms of pyoverdine-mediated virulence has only been developed very recently. In this study, we adapt a previously-established, murine macrophage-based model to use human bronchial epithelial (16HBE) cells. We demonstrate that conditioned medium from *P. aeruginosa* induced rapid 16HBE cell death through the pyoverdine-dependent secretion of cytotoxic rhamnolipids. Genetic or chemical disruption of pyoverdine biosynthesis decreased rhamnolipid production and mitigated cell death. Consistent with these observations, chemical depletion of lipids or genetic disruption of rhamnolipid biosynthesis abrogated the toxicity of the conditioned medium. Furthermore, we also examine the effects of exposure to purified pyoverdine on 16HBE cells. While pyoverdine accumulated within cells, it was largely sequestered within early endosomes, resulting in minimal cytotoxicity. More membrane-permeable iron chelators, such as the siderophore pyochelin, decreased epithelial cell viability and upregulated several pro-inflammatory genes. However, pyoverdine potentiated these iron chelators in activating pro-inflammatory pathways. Altogether, these findings suggest that the siderophores pyoverdine and pyochelin play distinct roles in virulence during acute *P. aeruginosa* lung infection.

**IMPORTANCE**  Multidrug-resistant *Pseudomonas aeruginosa* is a versatile bacterium that frequently causes lung infections. This pathogen is life-threatening to mechanically-ventilated patients in intensive care units and is a debilitating burden for individuals with cystic fibrosis. However, the role of *P. aeruginosa* virulence factors and their regulation during infection are not fully understood. Previous murine lung infection studies have demonstrated that the production of siderophores (e.g., pyoverdine and pyochelin) is necessary for full *P. aeruginosa* virulence. In this report, we provide further mechanistic insight into this phenomenon. We characterize distinct and novel ways these siderophores contribute to virulence using an *in vitro* human lung epithelial cell culture model.

**KEYWORDS**  *Pseudomonas aeruginosa*, virulence, siderophores, lung epithelial cells, inflammation, pyoverdine, pyochelin, rhamnolipids

Multidrug-resistant *Pseudomonas aeruginosa* is one of the most common Gram-negative, respiratory pathogens. It infects mechanically-ventilated patients in intensive care units and those with cystic fibrosis (CF) or chronic obstructive pulmonary disease (1–5). This pathogen's intrinsic resistance to several classes of antibiotics and

Address correspondence to Natalia V. Kirienko, kirienko@rice.edu.

The authors declare no conflict of interest.

See the funding table on p. 18.

exceptional ability to form biofilms on medical devices and airway tissue pose serious challenges for medical intervention (6, 7). *P. aeruginosa* also actively deploys numerous virulence factors and toxins that damage host tissue, further affecting pulmonary function (8). Two of the major virulence factors produced by this pathogen are the siderophores pyoverdine and pyochelin.

Several studies have proposed possible mechanisms of siderophore-dependent virulence during *P. aeruginosa* lung infection (9–12). Both pyoverdine and pyochelin scavenge ferric iron, providing the pathogen with this essential micronutrient during infection. Of the two, pyoverdine exhibits an affinity for ferric iron that is orders of magnitude higher than pyochelin, and the former is distinctly able to chelate the metal from host ferroproteins such as transferrin and lactoferrin (13, 14). Iron acquisition during infection is critical for promoting bacterial growth and biofilm formation (15, 16), and *P. aeruginosa* mutants lacking various iron uptake systems exhibit virulence attenuation during murine lung infection (17). It is important to note that these iron uptake systems do not contribute equally; of the two siderophores, pyoverdine appears to play a more significant role in *P. aeruginosa* virulence (17).

Pyoverdine-mediated iron uptake further promotes *P. aeruginosa* virulence by derepressing the alternative sigma factor PvdS, which activates the transcription of several virulence genes such as those encoding the translational inhibitor exotoxin A, the exoprotease PrpL (protease IV), and pyoverdine biosynthetic enzymes (12, 18). Furthermore, we have recently used a *Caenorhabditis elegans* nematode model to demonstrate that pyoverdine directly chelates host iron, disrupting mitochondrial homeostasis (19–21). Pyoverdine's well-documented role in acute lung infection is likely mediated by a combination of these various pathogenic functions (17, 22–25).

Recently, we established the first reported *in vitro* cell culture model for pyoverdine-dependent virulence, where murine macrophages were treated with conditioned medium from *P. aeruginosa* grown in serum-free cell culture medium (26). Under these conditions, *P. aeruginosa* exhibited robust pyoverdine production, yet the siderophore was not required for bacterial growth (Fig. 1A and B; Fig. S1A and B), allowing for the study of pyoverdine's role in virulence. This pyoverdine-rich conditioned medium from wild-type *P. aeruginosa* PAO1 was cytotoxic toward murine macrophages, including murine alveolar macrophages (Fig. S1C); in clinical isolates, pyoverdine content in the conditioned medium positively correlated with cytotoxicity (26).

In this report, we adapted the *in vitro* pyoverdine virulence model to use human bronchial epithelial (16HBE) cells to examine the consequences of pyoverdine production (i.e., exposure to pyoverdine and/or pyoverdine-regulated virulence factors) during *P. aeruginosa* lung infection. Conditioned medium from *P. aeruginosa* caused acute cell death and severe damage to the epithelial monolayer in a pyoverdine-, but not pyochelin-, dependent manner. Interestingly, this damage did not require host iron chelation nor production of the two known pyoverdine-regulated toxins, exotoxin A or PrpL. Instead, pyoverdine production led to the secretion of cytotoxic rhamnolipids that have previously been shown to permeabilize host membranes (27). Consistent with this observation, chemical depletion of lipids or genetic disruption of rhamnolipid production was sufficient to abrogate toxicity from the conditioned medium on 16HBE cells. Importantly, the pyoverdine biosynthetic inhibitor 5-fluorocytosine (5-FC) effectively inhibited rhamnolipid production and mitigated *P. aeruginosa* virulence in two highly virulent clinical isolates. We also examined the effects of exposing 16HBE cells to purified pyoverdine alone. While pyoverdine accumulated within cells, the siderophore was largely sequestered within early endosomes, showing minimal cytotoxicity. More membrane-permeable iron chelators, such as pyochelin, decreased epithelial cell viability and upregulated several pro-inflammatory pathways. Pyoverdine potentiated these iron chelators in activating pro-inflammatory pathways. Altogether, these findings suggest that pyoverdine and pyochelin play distinct roles in virulence during acute *P. aeruginosa* lung infections.

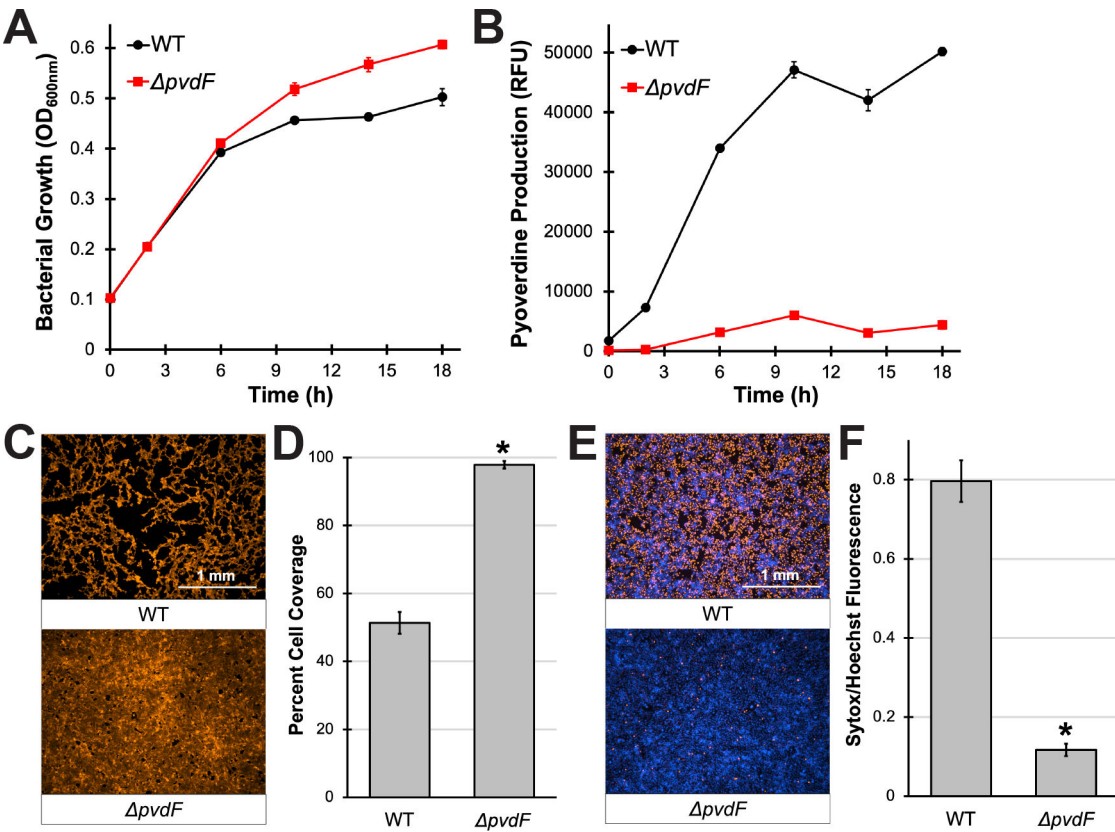

**FIG 1** Pyoverdine-rich conditioned medium kills 16HBE cells and damages the epithelial monolayer. (A and B) Bacterial growth (A) or pyoverdine production (B) of wild-type (WT) *P. aeruginosa* PAO1 or pyoverdine biosynthetic mutant (PAO1Δ*pvdF*) in serum-free Eagle's minimum essential medium (EMEM). (C) Fluorescent micrographs of 16HBE cells after 30-min exposure to conditioned medium from WT PAO1 or PAO1Δ*pvdF* grown in EMEM. Cells were prelabeled with CellMask Orange plasma membrane stain. (D) Quantification of percentage micrograph area covered by fluorescent cells. (E) Fluorescent micrographs of 16HBE cells after 15-min exposure to conditioned medium from WT PAO1 or PAO1Δ*pvdF* in the presence of Sytox Orange nucleic acid stain (red). Cells were prelabeled with Hoechst 33342 nucleic acid stain (blue). (F) Quantification of Sytox Orange mean fluorescence intensity normalized to that of Hoechst 33342. All error bars represent SEM from at least three biological replicates. *Corresponds to $P < 0.01$ based on Student's *t*-test.

## RESULTS

### Pyoverdine-rich, conditioned medium induces rapid cell death and damages the epithelial monolayer

To investigate the role of pyoverdine production during *P. aeruginosa* lung infection, we treated 16HBE cells with bacteria-free, pyoverdine-rich, conditioned medium from *P. aeruginosa* PAO1 grown in serum-free cell growth medium [Eagle's minimum essential medium (EMEM)]. To visualize the integrity of the epithelial monolayer, we prelabeled cells with a CellMask plasma membrane stain. Within 30 min, the conditioned medium severely damaged the monolayer, causing detachment of more than half of the cells (Fig. 1C and D). This disruption was significantly attenuated in 16HBE cells treated with identically prepared material from an isogenic pyoverdine biosynthetic mutant (PAO1Δ*pvdF*) (Fig. 1C and D). However, a pyochelin mutant (PAO1Δ*pchBA*) was indistinguishable from the wild-type bacteria (Fig. S1D and E). Preventing the biosynthesis of both pyoverdine and pyochelin (PAO1Δ*pvdF*Δ*pchBA*) conferred no more protection to 16HBE cells than disrupting pyoverdine alone (Fig. S1D and E). To determine whether cell detachment was caused by cell death (rather than from degradation of the extracellular matrix via bacterial proteases and other factors, for example), cells were labeled with cell-permeant (Hoechst 33342; labels all cells) and cell-impermeant (Sytox Orange; labels only dead cells) nucleic acid stains. Exposure to pyoverdine-rich, conditioned medium from wild-type bacteria caused rapid membrane permeabilization (within 15 min) and

internalization of the cell-impermeant nucleic acid stain, suggesting that the cells had died (Fig. 1E and F). In contrast, the conditioned medium from the pyoverdine mutant exhibited substantially lower cytotoxicity (Fig. 1E and F).

## 5-Fluorocytosine inhibits pyoverdine-mediated damage in highly virulent *P. aeruginosa* clinical isolates

Since genetic disruption of pyoverdine biosynthesis decreased the toxicity of the conditioned medium, we hypothesized that the same result could be accomplished using a chemical inhibitor. To that end, we tested whether the FDA-approved antimycotic and pyoverdine biosynthetic inhibitor 5-fluorocytosine (5-FC) (23, 28, 29) inhibited pyoverdine-dependent virulence. Several *P. aeruginosa* strains, including PAO1 and two clinical strains isolated from pediatric CF patients, PA2-72 and PA2-61, were selected for testing. These isolates were chosen from a large collection of CF isolates (22) for their high *in vitro* pyoverdine production and virulence against the nematode host *C. elegans* (Fig. S2). These isolates also exhibited substantial *in vivo* pyoverdine production and host mortality during acute murine lung infection (22). 5-FC significantly impaired pyoverdine production in PAO1 and PA2-72 when these strains were grown in EMEM, without overtly affecting bacterial growth (Fig. 2A through C). 5-FC reduced pyoverdine production in PA2-61 as well (Fig. 2C), although it pushed cells into an aggregated phenotype (Fig. 2B), confounding bacterial growth measurements by optical density. 5-FC also significantly attenuated 16HBE cell detachment and death after exposure to the conditioned medium from each of the three strains (Fig. 2D through G). These findings are consistent with previous work, where the inhibition of pyoverdine biosynthesis by 5-FC was sufficient to rescue invertebrate and mammalian hosts from *P. aeruginosa* virulence (22, 23, 30).

## Conditioned medium toxicity is mediated by secreted lipid factors

One key difference between pyoverdine and pyochelin is their affinity for ferric iron. Due to an exceptionally high affinity for the metal, pyoverdine is uniquely able to remove iron from host ferroproteins (13, 14) and induce a lethal hypoxic response in a *C. elegans* nematode model (19). We thus examined whether chelation of host iron was important for epithelial monolayer cell death. To hinder pyoverdine's ability to bind iron, we pretreated the pyoverdine-rich conditioned medium with gallium ($Ga^{3+}$) or ferric iron ($Fe^{3+}$), either of which would be bound by the siderophore. Gallium (III) has an ionic radius that is nearly identical to iron and binds nearly as tightly to the siderophore as ferric iron, but it cannot be dissociated by reduction the way that ferric iron can. Chelating either metal would be expected to prevent pyoverdine from scavenging iron from human epithelial cells. Surprisingly, even the addition of excess metal did not significantly reduce pathogenesis (Fig. 3A and B; Fig. S3), suggesting that death was not due to siderophore-mediated removal of host iron.

We also examined whether cytotoxicity from the conditioned medium could be attributed to pyocyanin content, since pyocyanin is known to cause acute oxidative damage to host cells (31–33). We observed that the pyoverdine biosynthetic mutant produced significantly less pyocyanin (~3.5 µM compared to ~6 µM for wild-type) in EMEM (Fig. S4A). However, 16HBE cells treated with commercially-sourced, purified pyocyanin at concentrations of either 6 or 60 µM failed to exhibit significant cell death within the previously observed timeframe (Fig. S4B and C). This indicated that pyocyanin was not responsible for the cytopathology.

Interestingly, we observed that the material responsible for cell death had a high molecular weight; removing material with a molecular mass greater than 10 kDa (via centrifugal filtration) virtually eradicated damage to the epithelial monolayer (Fig. 3A and B; Fig. S3), suggesting that a large macromolecule or molecular complex was responsible for the cytotoxicity of conditioned media. Both pyoverdine and pyocyanin are considerably smaller than this molecular weight threshold and thus are unlikely to be directly causing the cytotoxicity observed. This was consistent with experiments described above where iron and gallium supplementation had no effect on cell death.

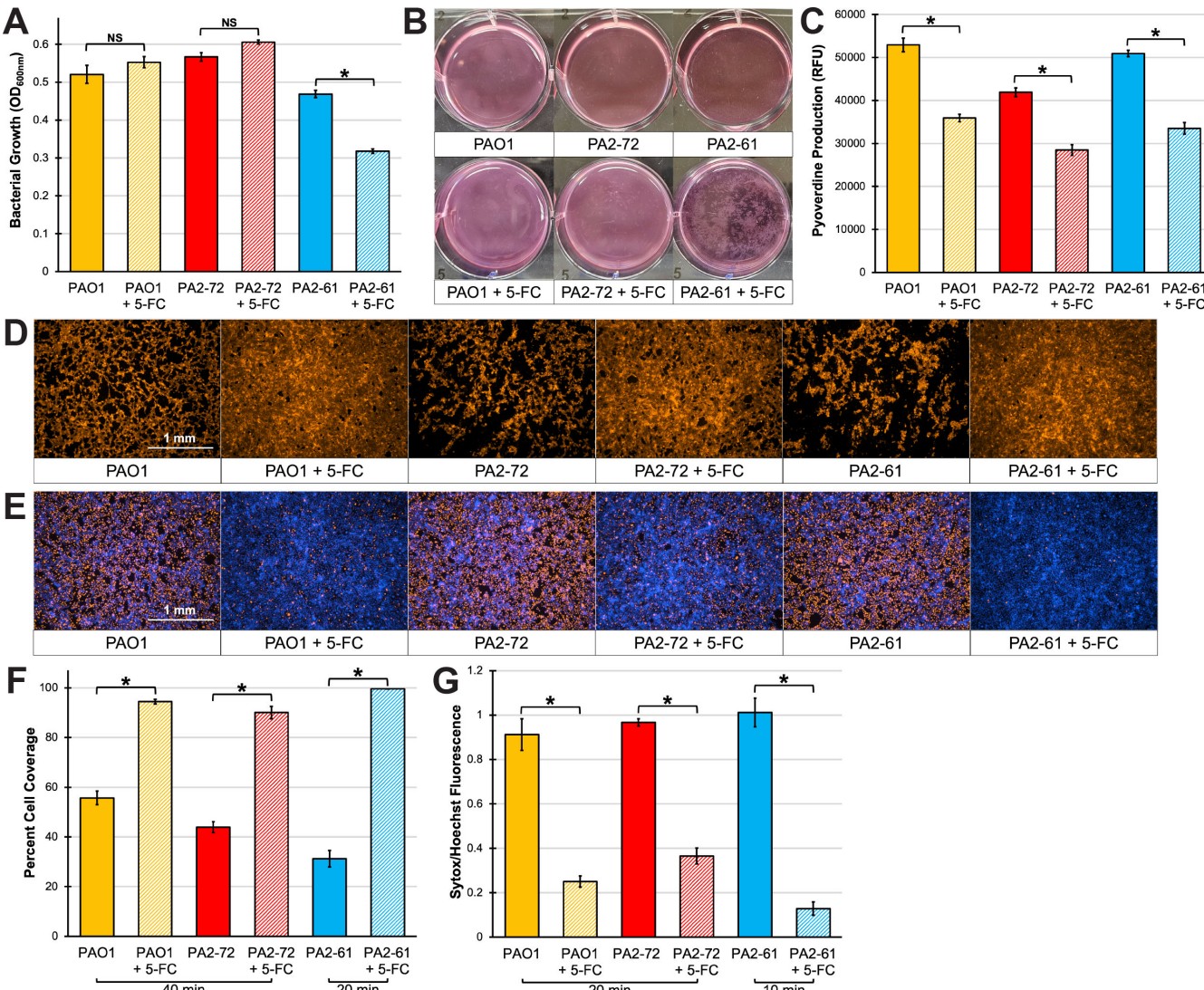

FIG 2   5-FC mitigates conditioned medium toxicity in highly-virulent cystic fibrosis isolates. (A) Bacterial growth of PAO1, PA2-72, or PA2-61 in EMEM with or without 100 µM 5-FC. (B) Photograph of EMEM culture after 18 h incubation. (C) Pyoverdine production of PAO1, PA2-72, or PA2-61 in EMEM with or without 100 µM 5-FC. (D) Fluorescent micrographs of 16HBE cells after exposure to conditioned EMEM for 40 min (PAO1 and PA2-72) or 20 min (PA2-61). Cells were prelabeled with CellMask Orange plasma membrane stain. (E) Fluorescent micrographs of 16HBE cells after exposure to conditioned EMEM supplemented with Sytox Orange nucleic acid stain (red) for 20 min (PAO1 and PA2-72) or 10 min (PA2-61). Cells were prelabeled with Hoechst 33342 nucleic acid stain (blue). (F) Quantification of percentage micrograph area covered by fluorescent cells in (D). (G) Quantification of Sytox Orange mean fluorescence intensity normalized to that of Hoechst 33342 in (E). All error bars represent SEM from four biological replicates. *Corresponds to $P < 0.01$ and NS corresponds to $P > 0.05$ based on a one-way analysis of variance (ANOVA) with Sidak's multiple comparisons test.

To investigate whether this material is proteinaceous, the conditioned medium was pretreated with proteinase K for 24 h at room temperature. Proteolytic digestion did not significantly attenuate the cytotoxicity of the conditioned medium (Fig. 3A and B). We were unable to determine the effects of pretreating conditioned meua with proteinase K on cell detachment; the treatment alone caused considerable damage to the extracellular matrix, making it very difficult to unambiguously assign contributions from the proteinase compared to the materials in the medium released by the bacteria (Fig. S3). However, genetically disrupting the two pyoverdine-regulated toxins, exotoxin A or PrpL, or the type II secretion system through which they are secreted, did not significantly alter cytotoxicity (Fig. S5), which was consistent with proteinase K observations.

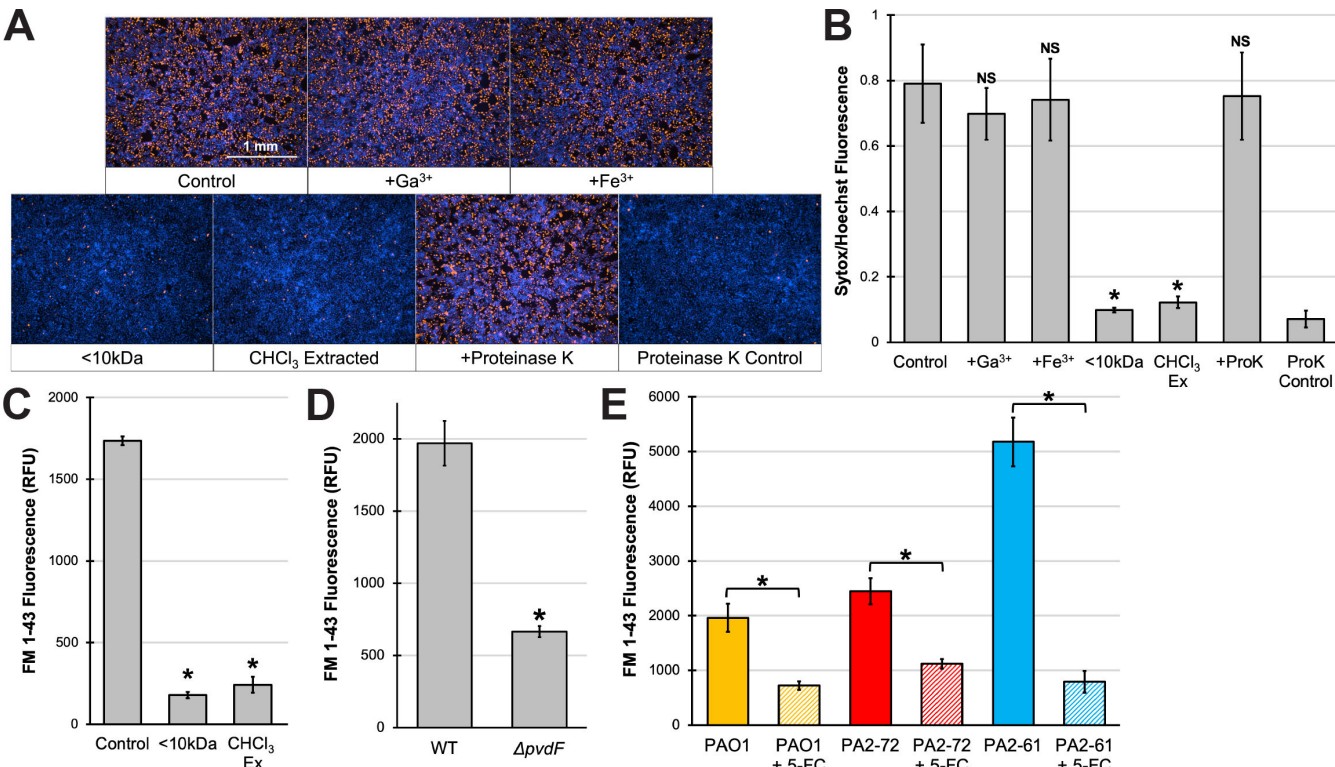

**FIG 3** Lipid factors drive pyoverdine-rich conditioned medium toxicity. (A) Fluorescent micrographs of 16HBE cells after 15-min exposure to conditioned medium from WT PAO1 in the presence of Sytox Orange nucleic acid stain (red). Conditioned medium was pretreated with 200 µM Ga(NO₃)₃, 200 µM FeCl₃, or 100 µg/mL proteinase K for 24 h, had macromolecules depleted via a 10-kDa centrifugal filter, or had lipids depleted by chloroform (CHCl₃) extraction. Cells were prelabeled with Hoechst 33342 nucleic acid stain (blue). (B) Quantification of Sytox Orange mean fluorescence intensity normalized to that of Hoechst 33342. (C) Quantification of lipids by FM 1-43 fluorescent labeling in conditioned medium from WT PAO1 (control) or macromolecule- (<10 kDa) or lipid- (CHCl₃) depleted material. (D) Quantification of lipids in conditioned medium from WT PAO1 or PAO1Δ*pvdF* by FM 1-43 labeling. (E) Lipid production by PAO1, PA2-72, or PA2-61 in EMEM with or without 100 µM 5-FC. All error bars represent SEM from at least three biological replicates. *Corresponds to *P* < 0.01 and NS corresponds to *P* > 0.05 based on Student's *t*-test (D) or a one-way ANOVA with Dunnett's multiple comparisons test (B and C) or Sidak's multiple comparisons test (E).

In contrast, chloroform-mediated extraction of lipids from conditioned media abrogated cytotoxicity and limited damage to the epithelial monolayer (Fig. 3A and B; Fig. S3). Based on these results, we used the lipophilic dye FM 1-43 to directly measure lipid content in conditioned medium. This probe is nonfluorescent in an aqueous solution but becomes highly fluorescent upon binding lipid membranes (34). Both chloroform extraction and centrifugal filtration of macromolecules (>10 kDa) significantly depleted FM 1-43-mediated fluorescence, suggesting that each treatment removed a large portion of the lipid material from the medium (Fig. 3C). Interestingly, conditioned medium from the pyoverdine biosynthetic mutant had considerably less lipid material than wild-type PAO1 (Fig. 3D). This effect was recapitulated by treatment with 5-FC, which reduced the release of lipid factors from PAO1, PA2-61, or PA2-72 (Fig. 3E). Under basal conditions, conditioned medium from PA2-61 exhibited markedly higher lipid contents than the other strains (Fig. 3E), consistent with its higher toxicity against 16HBE cells (Fig. 2E and G).

## Pyoverdine regulates rhamnolipid production

Based on previous studies (27, 35), we posited that the relevant secreted lipid factors were rhamnolipids. We recently demonstrated that *P. aeruginosa* secretes rhamnolipids that rapidly induce membrane rupture and permeabilization in a wide range of host cells, including murine macrophages, human bronchial epithelial cells, and erythrocytes (27). To test this hypothesis, we measured lipid content in conditioned medium inoculated

with a rhamnolipid biosynthetic mutant, MPAO1*rhlA*. Conditioned medium from this mutant had greatly reduced FM 1-43 staining, reinforcing the conclusion that the marker was at least partially staining rhamnolipids. This decrease was likely specific to reduction in rhamnolipid production, as this strain maintained normal growth patterns and pyoverdine production, comparable to the control strain MPAO1*cat* (which has the transposon inserted in an extraneous gene encoding a chloramphenicol acetyltransferase Fig. 4A through C). Importantly, the conditioned medium from MPAO1*rhlA* neither damaged the 16HBE monolayer nor induced cell death (Fig. 4D through G). This outcome was anticipated, since rhamnolipid production was compromised.

We observed similar results (i.e., reduced lipid content and lower toxicity) for mutants with transposons inserted into genes encoding the RhlRI quorum sensing system (MPAO1*rhlR* and MPAO1*rhlI*) (Fig. 4A through G), suggesting that this pathway regulates rhamnolipid production during growth in EMEM (36). The best characterized function of the *rhlI and rhlR* genes is the production and detection of quorum-sensing molecules, specifically *N*-butanoyl-L-homoserine lactone (C4-HSL) (37). To test whether quorum-sensing is impaired in pyoverdine biosynthetic mutants, we took advantage of an *Escherichia coli*-based bioluminescent reporter that responds to extracellular quorum-sensing molecules (38). Using this reporter, we quantified C4-HSL concentrations in the conditioned medium of pyoverdine mutants (PAO1Δ*pvdF*, MPAO1*pvdF*) and saw no significant difference in C4-HSL production between mutants and their pyoverdine-producing counterparts (WT PAO1 and MPAO1*cat*) (Fig. 4H; Fig. S6A through D). In contrast, conditioned medium from the MPAO1*rhlI* C4-HSL biosynthetic mutant was comparable to uninoculated medium (Fig. S6A through D), indicating that the reporter's response was selective and was specific to extracellular C4-HSL.

Another possibility was that pyoverdine regulated the transcription of the *rhlR* and *rhlI* genes. However, pyoverdine biosynthetic mutants did not exhibit a substantial (fold change ≥|1.5|) decrease in *rhlI* or *rhlR* expression (Fig. S6E). These results imply that if pyoverdine was to regulate rhamnolipid production through the Rhl pathway, it would likely be through a downstream target or mechanism parallel to RhlR-binding C4-HSL. One obvious possibility was that pyoverdine regulated the expression of *rhlA* or *rhlB*, which are genes directly responsible for rhamnolipid biosynthesis. However, *rhlA* and *rhlB* expression was unchanged in the pyoverdine mutants (Fig. S6E), suggesting that pyoverdine regulates rhamnolipid production through other means, possibly by altering its egress from cells.

## Rhamnolipids induce 16HBE cell death and affect *P. aeruginosa* swarming motility

Next, we examined the effects of exposing 16HBE cells to purified rhamnolipids (27). At concentrations comparable to those seen in conditioned medium from wild-type PAO1 or MPAO1*cat* (Fig. 5A; Fig. S7A), purified rhamnolipids killed 16HBE cells (Fig. 5B and C; Fig. S7C and E). However, purified rhamnolipids caused less detachment of cells than pyoverdine-rich conditioned medium (Fig. S7B and D), suggesting that other secreted factors (e.g., proteases) were at least partially responsible for the damage to the epithelial monolayer. Consistent with this interpretation, heat denaturation of the conditioned medium did not affect rhamnolipid content (Fig. S7F) or cytotoxicity of the material (Fig. S7H and J), but it did reduce cell detachment (Fig. S7G and I). Since the type II secretion system is responsible for the secretion of several *P. aeruginosa* protein toxins and the majority of secreted proteases (Fig. S5E) (39), we hypothesized that the factor causing detachment was likely to be secreted by this system. In contrast to this, treatment of 16HBE cells with conditioned medium from a mutant lacking the outer membrane transporter of the type II secretion system (MPAO1*xcpQ*) still caused substantial cell detachment (Fig. S5A and C). One caveat of this observation was that while the disruption of *xcpQ* substantially impaired protease secretion, it was not completely abolished (Fig. S5E). The likeliest explanation for these observations is that the protease(s) involved

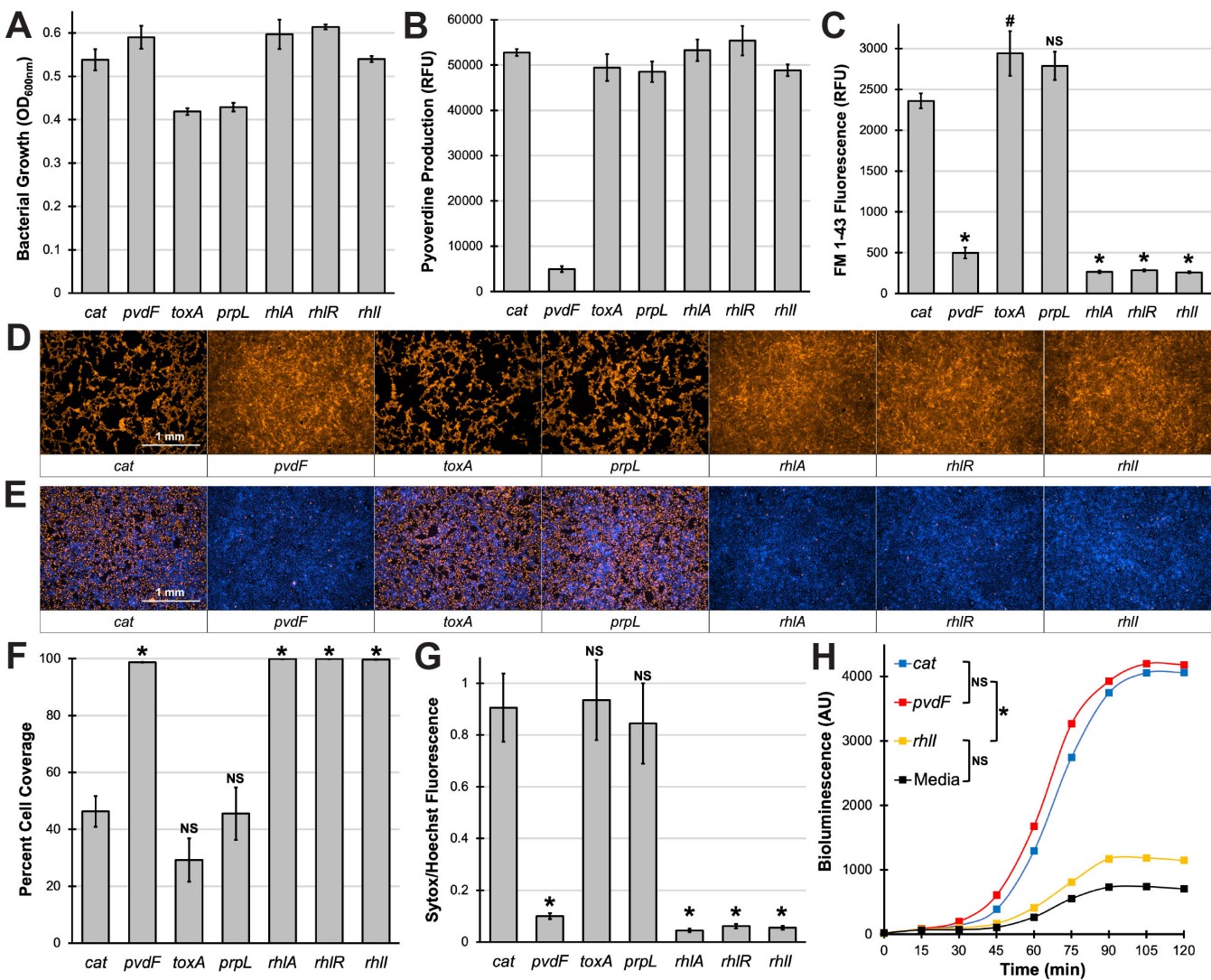

**FIG 4** Pyoverdine regulates the production of rhamnolipids. (A and B) Bacterial growth (A) or pyoverdine production (B) by MPAO1 transposon mutants in serum-free EMEM. (C) Quantification of FM 1-43 fluorescent labeling of lipids in conditioned medium from MPAO1 transposon mutants. (D) Fluorescent micrographs of 16HBE cells after exposure to conditioned medium from MPAO1 transposon mutants for 30 min. Cells were prelabeled with CellMask Orange plasma membrane stain. (E) Fluorescent micrographs of 16HBE cells after exposure to conditioned medium from MPAO1 transposon mutants in the presence of Sytox Orange nucleic acid stain (red) for 15 min. Cells were prelabeled with Hoechst 33342 nucleic acid stain (blue). (F) Quantification of percentage micrograph area covered by fluorescent cells in (D). (G) Quantification of Sytox Orange mean fluorescence intensity normalized to that of Hoechst 33342 in (E). (H) Bioluminescence produced by an *N*-butanoyl-L-homoserine lactone reporter strain (*Escherichia coli* JM109 pSB536) grown in media supplemented with conditioned medium from MPAO1 transposon mutants. All error bars represent SEM from three biological replicates. *Corresponds to *P* < 0.01, #corresponds to *P* < 0.05, and NS corresponds to *P* > 0.05 based on a one-way ANOVA with Dunnett's multiple comparisons test (C, F, G) or Tukey's multiple comparisons test (H) – see Fig. S6.

in epithelial cell detachment were still being produced and secreted by some other mechanism.

In addition to killing host cells, rhamnolipids are known to regulate *P. aeruginosa* swarming motility, which promotes pathogen proliferation and biofilm formation within the host (40–42). To test whether pyoverdine production affects swarming motility, we measured lawn growth on semisolid EMEM agar (0.5%) for wild-type PAO1, the pyoverdine biosynthetic mutant Δ*pvdF*, and the rhamnolipid biosynthetic mutant *rhlA*. As expected, swarming motility was diminished in the pyoverdine mutant and further

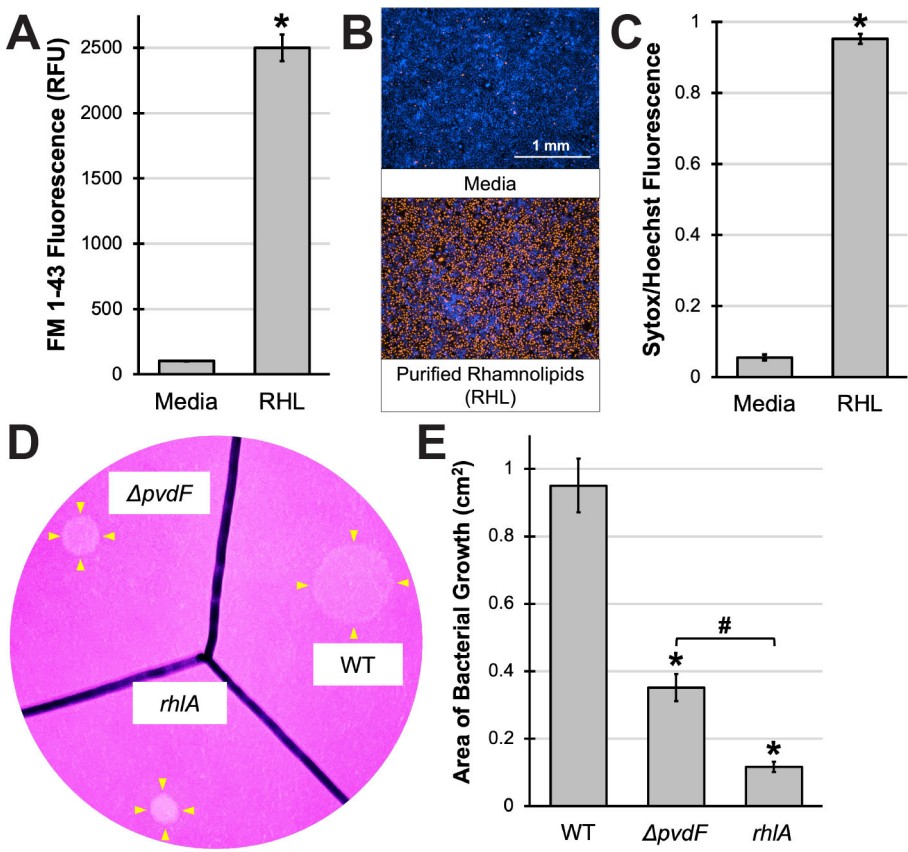

**FIG 5** Rhamnolipids mediate 16HBE cell death and *P. aeruginosa* swarming motility. (A) Quantification of FM 1-43-labeled lipids in EMEM supplemented with purified rhamnolipids (RHL). (B) Fluorescent micrographs of 16HBE cells after exposure to purified rhamnolipids from *P. aeruginosa* in the presence of Sytox Orange nucleic acid stain (red) for 15 min. Cells were prelabeled with Hoechst 33342 nucleic acid stain (blue). (C) Quantification of Sytox Orange mean fluorescence intensity normalized to that of Hoechst 33342 in (B). (D) Photograph of WT PAO1, PAO1*ΔpvdF*, or MPAO1*rhlA* growth on semisolid EMEM agar. (E) Quantification of area of bacterial growth for each strain in (D). All error bars represent SEM from three biological replicates. *Corresponds to $P < 0.01$ and #corresponds to $P < 0.05$ based on Student's *t*-test (A and C) or a one-way ANOVA with Tukey's multiple comparisons test (E).

reduced in the rhamnolipid mutant (Fig. 5D and E), suggesting that pyoverdine-mediated regulation of rhamnolipid production plays multiple roles in *P. aeruginosa* virulence.

## Pyoverdine translocates into 16HBE cells but is sequestered in early endosomes

Next, we wanted to investigate the consequences of exposing 16HBE cells to pyoverdine in the absence of other virulence factors. To this end, we developed a method for purifying pyoverdine from a pyoverdine-rich, bacteria-free conditioned medium. In brief, pyoverdine-rich bacterial filtrate was subjected to two purification steps: adsorption chromatography and reverse-phase HPLC (Fig. 6A through C). We tested whether pyoverdine purified in this way was toxic to 16HBE cells using a resazurin-based cell viability assay. We also compared its toxicity to several other iron-chelating compounds, the ferric iron chelator ciclopirox olamine, the ferrous iron chelator 1,10-phenanthroline, or the siderophores pyochelin (from *P. aeruginosa*) or deferoxamine (from *Streptomyces* spp.). Although the other iron chelators exhibited time- and dose-dependent cytotoxicity toward 16HBE cells, pyoverdine was largely nontoxic (Fig. 6D) even after 72 h treatment at 200 µM (Fig. S8A and B).

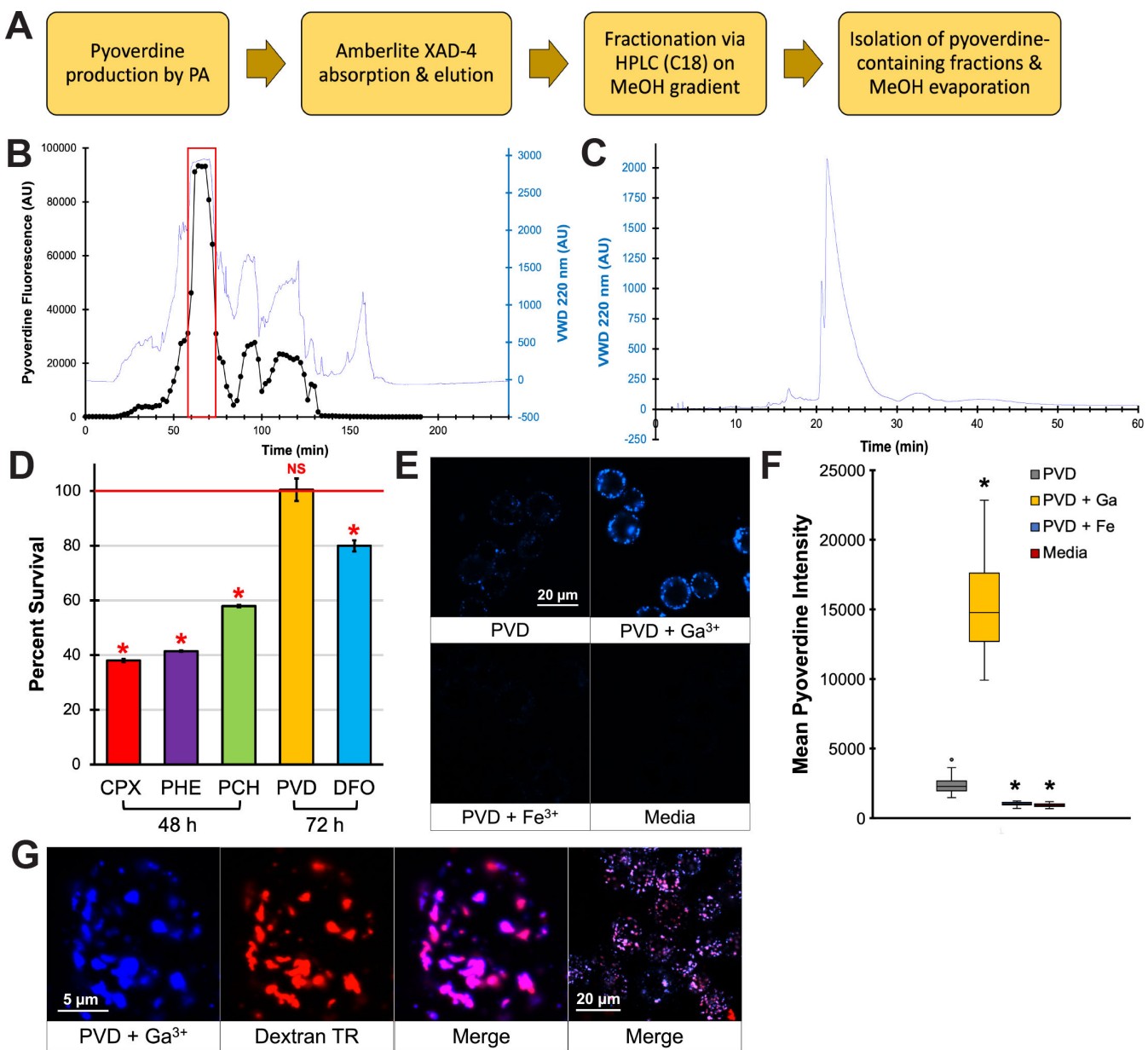

**FIG 6** Pyoverdine translocates into 16HBE cells and localizes to early endosomes. (A) Summary of the pyoverdine purification pipeline. (B) Representative chromatogram from the HPLC purification step of the pipeline. The red box depicts the predominant pyoverdine-containing fractions that were collected. (C) Analysis of the final purified product via HPLC. (D) 16HBE cell viability after 48 h treatment with 100 µM ciclopirox olamine, 1,10-phenanthroline, or pyochelin, or 72 h treatment of 100 µM pyoverdine or deferoxamine in serum-free EMEM. (E) Confocal micrographs of 16HBE cells exposed to 100 µM purified pyoverdine, pyoverdine with excess Ga(NO$_3$)$_3$, pyoverdine with excess FeCl$_3$, or media control for 24 h. Cells were trypsinized prior to imaging. (F) Quantification of pyoverdine fluorescence within 30 individual cells. (G) Confocal micrographs of 16HBE cells treated with 100 µM pyoverdine–gallium and Texas Red-labelled Dextran (10,000 molecular weight (MW)). Error bars in (D) represent SEM from three biological replicates. Error bars in (C) represent SD. *Corresponds to $P < 0.01$ based on a one-way ANOVA with Dunnett's multiple comparisons test.

Pyoverdine, with a molecular weight (MW) of ~1,365 g/mol, is considerably larger than these other iron chelators. We hypothesized that this size may limit the ability of the molecule to translocate across cell membranes. We took advantage of pyoverdine's intrinsic spectral properties to examine whether pyoverdine could get into 16HBE cells. After 24 h, considerable amounts of pyoverdine had entered 16HBE cells (Fig. 6E and F). Consistent with previous studies (21, 26), intracellular fluorescence was increased when pyoverdine was pre-saturated with gallium and was quenched when pyoverdine was

pre-saturated with iron (Fig. 6E and F). Importantly, pyoverdine fluorescence did not colocalize with CellMask, which labels the plasma membrane. Instead, confocal microscopy demonstrated that pyoverdine fluorescence was located within cells (Fig. S8C); pyoverdine fluorescence formed distinct punctae within the cell. Based on previous observations in murine macrophages (26), we hypothesized that pyoverdine was being sequestered within early endosomes. Supporting this hypothesis, pyoverdine colocalized with fluorophore-conjugated 10-kDa dextran, a well-established marker of the endosome (Fig. 6G; Fig. S8D) (43). Observed pyoverdine sequestration is consistent with our findings that pyoverdine, unlike other iron-chelating molecules, exhibited low cytotoxicity toward 16HBE cells.

## Iron chelation activates a pro-inflammatory response in 16HBE cells

While pyochelin exhibits a lower affinity toward ferric iron than pyoverdine, it is also substantially smaller, with a molecular weight of ~325 g/mol. We hypothesized that pyochelin may be able to enter 16HBE cells and chelate intracellular iron. While we were not able to visualize pyochelin within cells (due to its lack of distinct spectral properties), one likely consequence of iron removal in epithelial cells would be the activation of a pro-inflammatory transcriptional response. Several studies have demonstrated that iron chelation by various siderophores such as deferoxamine or enterobactin stimulates the production of pro-inflammatory cytokines, most notably IL-8 in lung epithelial cells, intestinal epithelial cells, or oral keratinocytes (44–46). To reaffirm these findings, we treated 16HBE cells with various iron chelators and measured the mRNA levels of genes involved in neutrophilic inflammation. We first observed that total RNA yield (from phenol–chloroform extraction) in these cells was correlated with the resazurin-based cell viability assay (Fig. 6D). Cells treated with small-molecule (<1,000 MW), cytotoxic iron chelators (ciclopirox olamine, phenanthroline, pyochelin, or deferoxamine) yielded lower quantities of RNA, while cells treated with pyoverdine provided RNA quantities comparable to media controls (Fig. 7A). Using quantitative reverse transcription-PCR (qRT-PCR), we measured the expression of genes encoding *NLRP3* and *NLRP1*, components of the inflammasome, and those encoding several pro-inflammatory cytokines, including *IL1B*, *IL8*, and *TNF*. All of these genes have been associated with inflammation during lung infection (47). Except for pyoverdine, all iron chelators induced the expression of these pro-inflammatory genes (Fig. 7B). For *IL8*, we validated the qRT-PCR results via enzyme-linked immunosorbent assay (ELISA) to confirm that transcriptional activation led to increased cytokine production. Cells treated with cytotoxic iron chelators exhibited time-dependent increases in IL-8 secretion that correlated with *IL8* mRNA levels (Fig. 7C; Fig. S9). In contrast, cells treated with pyoverdine showed IL-8 secretion comparable to that of media control (Fig. 7C). We saw this transcriptional response in both wild-type 16HBE cells and in 16HBE cells carrying causative mutations in the cystic fibrosis transmembrane conductance regulator (CFTR G551D and CFTR ΔF508) (Fig. S10; online supplementary file 2) (48).

To ensure that the observed pro-inflammatory response was due to iron chelation, we presaturated pyochelin and deferoxamine with excess gallium (1:2 stoichiometric ratio) prior to exposure. Cells treated with gallium-bound pyochelin or deferoxamine yielded RNA quantities comparable to the media control, suggesting that gallium inhibited the cytotoxic effects of the siderophores (Fig. 7D). This pretreatment also significantly decreased the expression of pro-inflammatory genes (Fig. 7E), suggesting that the siderophore-induced inflammatory response was due to iron chelation rather than other nonspecific reactions or contaminants in the commercially-sourced material.

Finally, we investigated whether pyoverdine can indirectly promote lung inflammation by potentiating other iron-chelating molecules. Due to its exceptionally high affinity for iron, pyoverdine is likely to remove iron from other, more cell-permeable siderophores or to outcompete them for trace iron in the extracellular milieu. We suspect that this increases the pool of apo-siderophores that can promote inflammation. To test this hypothesis, we treated 16HBE cells with deferoxamine, pyoverdine, or both. Cells treated

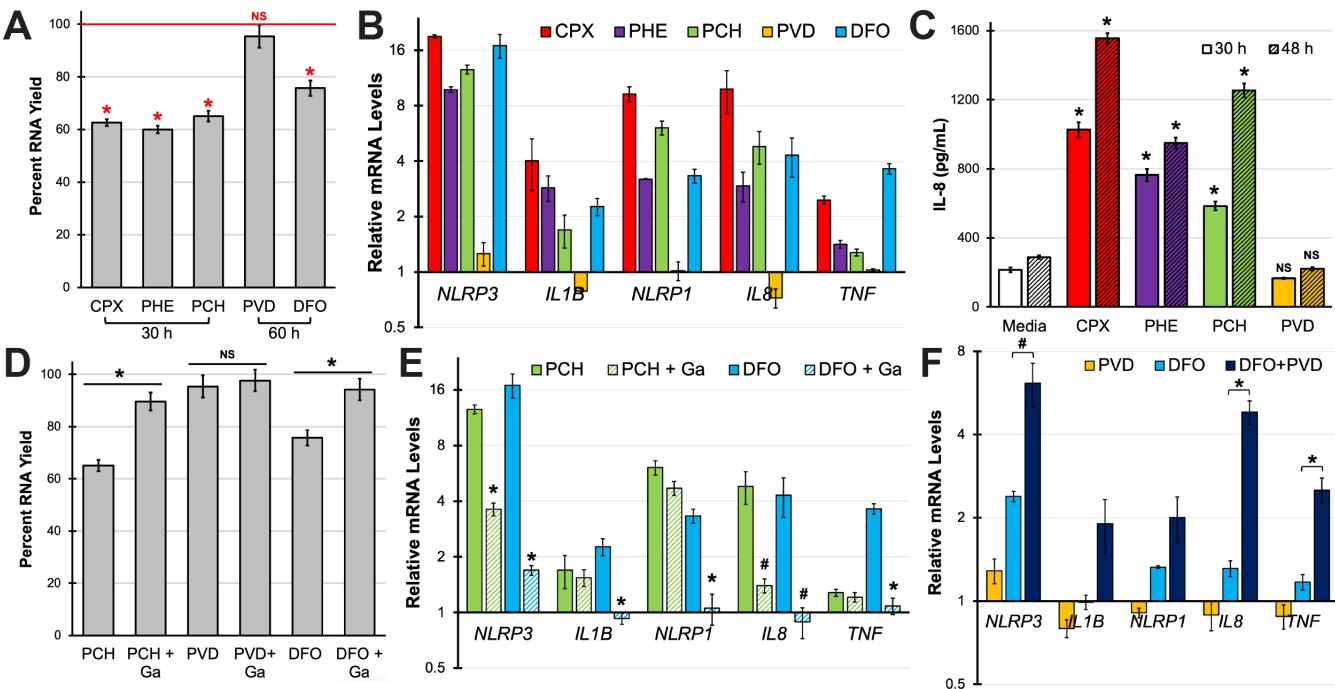

**FIG 7** Small-molecule iron chelators promote the expression of pro-inflammatory genes in 16HBE cells. (A) Total RNA yield in 16HBE cells treated with ciclopirox olamine (CPX), 1,10-phenanthroline (PHE), or pyochelin (PCH) for 30 h or cells treated with pyoverdine (PVD) or deferoxamine (DFO) for 60 h. RNA yield is shown normalized to media control. All treatments were at 100 µM in serum-free EMEM. (B) Pro-inflammatory gene expression (*NLRP3, IL1B, NLRP1*, *IL8,* or *TNF*) in cells treated with iron chelators or media control. Transcript levels were measured by qRT-PCR. (C) IL-8 protein concentration in the supernatants of 16HBE cells treated with iron chelators. IL-8 was quantified by ELISA. (D, E) Total RNA yield (D) or pro-inflammatory gene expression (E) in 16HBE cells treated with iron chelators with or without excess Ga(NO$_3$)$_3$ supplementation. (F) Pro-inflammatory gene expression after 60 h-treatment with pyoverdine, deferoxamine, or both molecules. All error bars represent SEM from three biological replicates. *Corresponds to $P < 0.01$, #corresponds to $P < 0.05$, and NS corresponds to $P > 0.05$ based on a one-way ANOVA with Dunnett's (A, C), Sidak's (D, E), or Tukey's (F) multiple comparisons test.

with both siderophores exhibited higher expression of pro-inflammatory genes compared to those treated with deferoxamine alone (Fig. 7F). Considering that pyoverdine alone did not affect the transcription of pro-inflammatory genes, these results suggest that pyoverdine enhanced the damage caused by deferoxamine by removing iron from deferoxamine, effectively increasing the pool of apo-deferoxamine.

## DISCUSSION

One of the greatest challenges to combating *P. aeruginosa* infections is the sheer multitude of virulence factors produced by the bacterium that contribute to pathogenesis. These include small-molecule virulence factors (e.g., siderophores and quorum-sensing molecules), factors involved in biofilm formation and motility (e.g., exopolysaccharides, type IV pili, and flagella), and more than 20 toxins that either directly kill host cells (e.g., exotoxin A, exoenzyme S, exotoxin T, and exotoxin U) or damage host tissue (e.g., elastase LasA, elastase LasB, PrpL, and alkaline protease) (37, 49–52). This complexity casts a pall over the prospects of epidemiological or therapeutic interventions.

The biomedical community hopes to eventually use molecular surveillance tools, such as whole-genome sequencing and mass spectrometry, to reliably predict a pathogen's ability to cause disease and then target treatments based on those pathogenic mechanisms using antivirulence. Due to the complexity of virulence in *P. aeruginosa*, this goal is only feasible with a more comprehensive understanding of the complicated interplay of these factors and by potentially targeting whole virulence networks rather than individual factors.

The results we report in this study suggest that the alternative sigma factor PvdS may be a promising target for therapeutic intervention during *P. aeruginosa* lung infections. PvdS regulates the production of several secreted toxins, including the translational inhibitor exotoxin A and the secreted protease PrpL. Exotoxin A, arguably one of the most extensively studied toxins in *P. aeruginosa*, inhibits protein synthesis (50, 53), inducing airway epithelial cell death (54) and inhibiting cell junction repair in the presence of *P. aeruginosa* elastase (55). Exotoxin A also contributes to *P. aeruginosa* virulence in various murine infection models (56–58). PrpL degrades host defense factors, like surfactant proteins and IL-22, that contribute to lung innate immunity (59–61). PrpL has also been shown to directly contribute to *P. aeruginosa* virulence during ocular infections (62, 63).

PvdS is best known for its role in pyoverdine biosynthesis and is indispensable for the production of pyoverdine biosynthetic enzymes. In addition to scavenging trace iron in the environment or directly from host ferroproteins, pyoverdine is involved in a positive feedback loop where the uptake of iron-bound pyoverdine by its outer membrane receptor, FpvR, derepresses PvdS, increasing the production of pyoverdine, exotoxin A, and PrpL (12, 18). Here, we have demonstrated that pyoverdine also promotes the production of another extracellular product with ramifications for host cell viability, rhamnolipids (Fig. 8). Secreted rhamnolipids have been shown to assemble into micellar structures (64) that directly interact with host membranes, causing rapid membrane rupture and cell death (35, 65, 66).

Although the two-dimensional human lung epithelial cell model we utilized here has limitations, such as the lack of epithelial cell polarity and mucus production (both of which are key factors in lung epithelial host–pathogen biology), our discovery of the importance of rhamnolipids in *P. aeruginosa*-mediated epithelial damage is consistent with several studies that used more physiologically-relevant *in vitro* cell culture (67) or mammalian tissue models (68, 69). And our findings linking pyoverdine to the regulation of rhamnolipid production add another dimension to understanding the network of *P. aeruginosa* virulence factors. One caveat is that it remains unclear how pyoverdine regulates rhamnolipid production and whether this mechanism is linked to quorum sensing, the primary mode of rhamnolipid regulation in the bacterium. While certain studies suggest that *P. aeruginosa* quorum sensing affects pyoverdine production (70–74), an inverse relationship has yet to be explored.

In addition to regulating secreted toxins, pyoverdine may also indirectly contribute to inflammation by removing iron from other, more cell-permeable siderophores, such as pyochelin, deferoxamine, or enterobactin. The latter is particularly worrisome in the context of polymicrobial infections with *Enterobacteriaceae* such as the respiratory pathogen *Klebsiella pneumoniae* (75). Importantly, while *P. aeruginosa* may lose the ability to produce pyoverdine during lung infection due to the emergence of social cheaters or transition in iron-acquisition strategies during the switch to chronic infection regimes (76–78), several surveys of patient sputum samples and clinical isolates have revealed that a large fraction of strains retain substantial capacity for pyoverdine production (22, 79–81). This reinforces the idea that pyoverdine may be an important target for therapeutic intervention.

Fortunately, an FDA-approved drug is currently available to mitigate this concern. 5-FC, an antimycotic that inhibits *pvdS* expression in *P. aeruginosa* and attenuates virulence during murine lung infection, was first identified as a potential anti-Pseudomonal by Imperi and colleagues in a screen for small molecules that inhibit pyoverdine production (23). We independently identified a chemical analog of 5-FC, 5-fluorouracil—which also inhibits *pvdS*—in a small-molecule screen for compounds that rescue *C. elegans* from *P. aeruginosa* in a pyoverdine-dependent pathogenesis model (28, 82). We also recently reported that 5-FC synergizes with another FDA-approved drug, gallium nitrate, to inhibit *P. aeruginosa* growth and virulence against *C. elegans* (29). Our findings in this study suggest that in addition to its bactericidal and biofilm-inhibitory activities (83, 84), gallium may also function as an anti-inflammatory agent during lung infection by

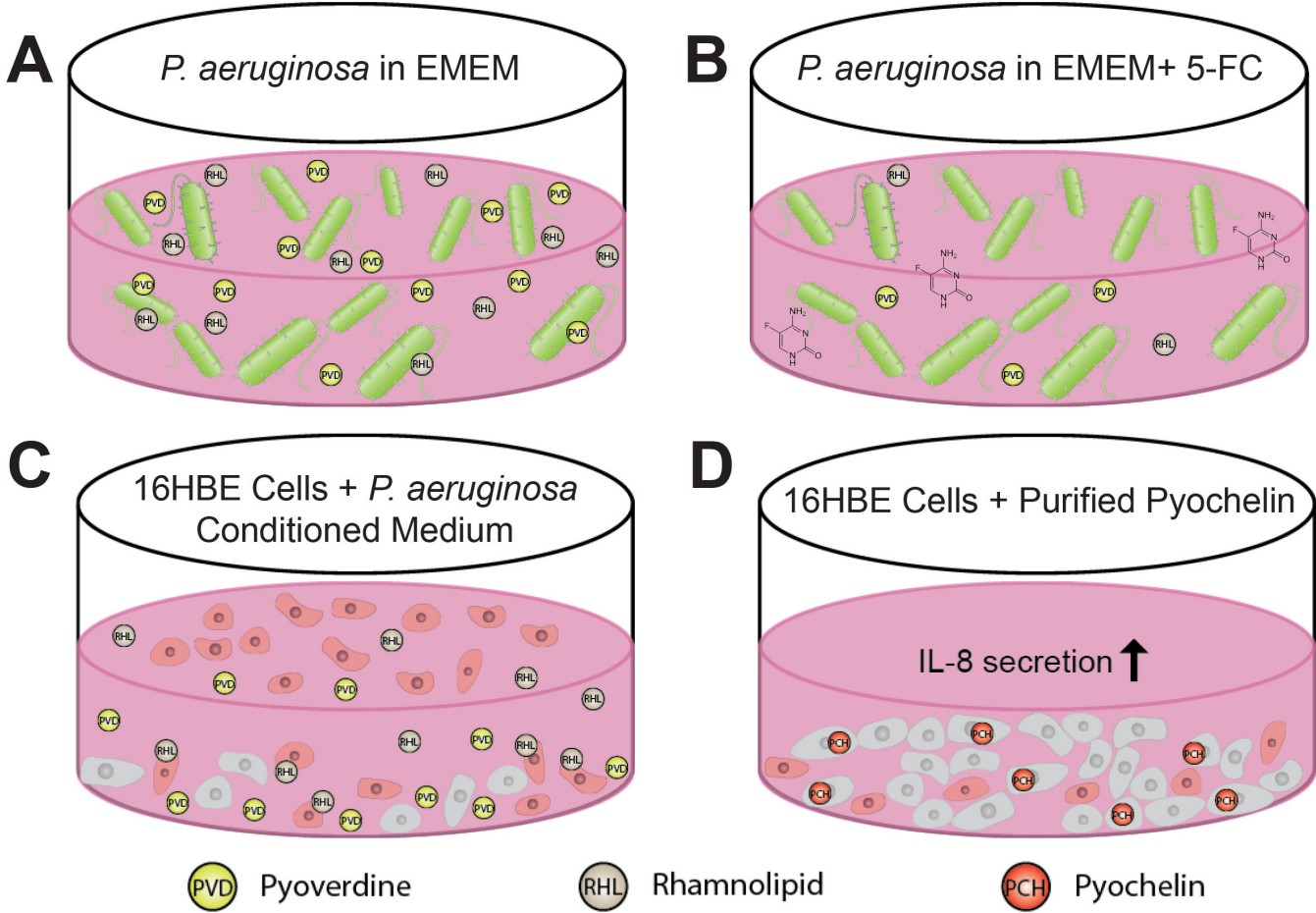

**FIG 8** *In vitro* lung epithelial cell model reveals novel roles for *Pseudomonas aeruginosa* siderophores. (A) Pyoverdine promotes rhamnolipid production by *P. aeruginosa* in EMEM. (B) The pyoverdine biosynthetic inhibitor 5-FC inhibits pyoverdine and rhamnolipid production in EMEM. (C) Pyoverdine- and rhamnolipid-rich *P. aeruginosa* conditioned medium rapidly induces 16HBE cell death. (D) Purified pyochelin, but not pyoverdine, decreases 16HBE cell viability and promotes the activation of pro-inflammatory responses, such as IL-8 secretion.

inhibiting intracellular iron chelation by pyochelin. This may mitigate not only epithelial cell death but also the profound activation of pro-inflammatory pathways that contribute to tissue damage (Fig. 8). These newly discovered roles for pyochelin further suggest that pyoverdine and pyochelin play distinct roles in *P. aeruginosa* virulence. Previous studies have shown that pyochelin triggers reactive oxygen species production in an iron-dependent manner in host cells during infection (85) or in other microbes during interbacterial competition (86–88). While pyochelin may not regulate additional virulence pathways, its ability to cross membranes makes this siderophore a potentially important mediator of host cell damage.

The benefits of suppressing pyochelin-mediated neutrophilic inflammation during lung infection, particularly chronic lung infection, have been well documented. Neutrophil-mediated mechanisms for bacterial clearance, such as the production of elastases, are important for host defense, but they also cause tissue damage by degrading extracellular matrix proteins (89, 90). During chronic infections (such as those in CF patients), these host defense factors continue to cause airway damage while the pathogen persists, exacerbating the decline in pulmonary function (91). While lung inflammation in CF patients is mediated by many factors, one of the most damaging is excess activation of the NLRP3 inflammasome. For this reason, investigators have sought methods to inhibit the NLRP3 inflammasome, such as MCC950. Recent investigations have shown promising results in murine infection studies (92, 93). Importantly,

the responses we observed to intracellular iron-chelation in lung epithelial cells (e.g., transcriptional upregulation of *NLRP3* and *IL1B*) have been associated with NLRP3 inflammasome priming (94). While gallium has been broadly associated with anti-inflammatory properties (95–97), studies have yet to specifically explore gallium's role in inhibiting pathogen-associated inflammation. Considering recent findings that bacterial siderophores promote inflammation (44–46, 75), this therapeutic avenue may merit consideration.

## MATERIALS AND METHODS

### Bacterial strains and growth conditions

See the list of bacterial strains given in Table 1. All MPAO1 transposon insertion sites were verified by Sanger sequencing (98). Insertions were determined by adapting a previously established method for the MAR2xT7 transposon library in PA14 (99) and primers are in Table S1. To produce pyoverdine-rich conditioned medium, an LB (Luria broth) overnight culture of *P. aeruginosa* was diluted 20-fold into 2 mL of serum-free EMEM (MilliporeSigma, St. Louis, MO, USA) in a six-well plate. The plate was sealed with a Breathe-Easy sealing membrane (Diversified Biotech, Dedham, MA, USA) and grown statically at 37°C for 18 h. Pyoverdine production (Ex. 405 nm; Em. 460 nm) and bacterial growth (Abs. 600 nm) were measured spectrophotometrically on a Cytation5 Multimode Reader (Biotek, Winnoski, VT, USA). Bacteria were then removed by centrifugation and the supernatant was treated with an antibiotic combination to kill residual bacteria (100 µg/mL amikacin, 100 µg/mL gentamicin, and 100 µg/mL tobramycin).

### Cell culture

Wild-type and mutant 16HBE cells (Table 1) were passaged in EMEM supplemented with 10% fetal bovine serum (Corning, Corning, NY, USA), penicillin/streptomycin (MilliporeSigma), and MEM non-essential amino acids (MilliporeSigma).

For experiments with *P. aeruginosa* conditioned medium, $4 \times 10^6$ cells were seeded into each well of a collagen (type I from calf skin - MilliporeSigma)-coated 12-well plate and grown at 37°C for ~24 h in a $CO_2$-jacketed incubator until they reached 100% confluence. To visualize the epithelial monolayer, cells were stained with 2.5 µg/mL CellMask Orange plasma membrane stain (Invitrogen, Carlsbad, CA, USA) for 1 h prior to conditioned medium exposure. Following the treatment, the medium was aspirated, and the monolayer was imaged on a Cytation5 Multimode Reader using an RFP filter cube. The percentage image area covered by fluorescent cells was quantified using ImageJ. To visualize cell death, cells were prelabeled with 20 µM Hoechst 33342 (ThermoFisher Scientific, Waltham, MA, USA) for 30 min and then exposed to a conditioned medium in the presence of 2.5 µM Sytox Orange (Invitrogen). Following the treatment, the medium was aspirated, and the monolayer was imaged on a Cytation5 Multimode Reader using DAPI (for Hoechst 33342) and RFP (for Sytox Orange) filter cubes. Images were exported and quantified for mean blue or red fluorescence intensity on ZEN Blue image analysis software (Zeiss, Oberkochen, Germany).

For 16HBE cell viability measurements following the iron chelator treatment, 440 µM resazurin (ThermoFisher Scientific) in phosphate-buffered saline was diluted 10-fold into the treatment medium, and cells were incubated for 1.5 h. The medium was collected and briefly centrifuged to remove cells. One hundred fifty microliters of the supernatant were transferred to a 96-well plate, and resorufin (reduced resazurin) fluorescence (Ex. 560 nm; Em. 590 nm) was measured on a Cytation5 Multimode Reader.

### Pyocyanin measurement

Pyocyanin in the *P. aeruginosa* EMEM-conditioned medium was quantified as previously published (100). In brief, pyocyanin was extracted in chloroform. Pyocyanin in

**TABLE 1** List of bacterial strains and cell lines used in this study

| Strains | Relevant genotype | Source or reference |
|---|---|---|
| *Pseudomonas aeruginosa strains* | | |
| PAO1 | WT | D. Haas |
| PAO1Δ*pvdF* | Δ*pvdF* | D. Haas |
| PAO1Δ*pchBA* | Δ*pchBA* | D. Haas |
| PAO1Δ*pvdFpchBA* | Δ*pvdFpchBA* | D. Haas |
| MPAO1 PW2280 | *cat* (ISlacZ/hah) | (98) |
| MPAO1 PW5033 | *pvdF* (ISlacZ/hah) | (98) |
| MPAO1 PW3078 | *toxA* (ISphoA/hah) | (98) |
| MPAO1 PW8077 | *prpL* (ISphoA/hah) | (98) |
| MPAO1 PW6887 | *rhlA* (ISphoA/hah) | (98) |
| MPAO1 PW6882 | *rhlR* (ISlacZ/hah) | (98) |
| MPAO1 PW6880 | *rhlI* (ISphoA/hah) | (98) |
| MPAO1 PW5085 | *pvdS* (ISlacZ/hah) | (98) |
| MPAO1 PW6223 | *xcpQ* (ISphoA/hah) | (98) |
| PA2-72 | WT (CF isolate) | (22) |
| PA2-61 | WT (CF isolate) | (22) |
| *Other bacterial strains* | | |
| *Escherichia coli* JM109 pSB536 | *Pahyl::luxABCDE* regulated by constitutive AhyR | (38) |
| *Human bronchial epithelial cells* | | |
| 16HBE | WT CFTR | G. Bao |
| 16HBE G551D | CFTR G551D | G. Bao |
| 16HBE ΔF508 | CFTR ΔF508 | G. Bao |

the organic layer was acidified and solubilized in 0.2 M hydrochloric acid. Pyocyanin concentration in the HCl solution was quantified by absorbance (Abs. 520 nm) on a Cytation5 Multimode Reader and calculated using a standard curve based on identically prepared EMEM standards supplemented with commercially sourced pyocyanin (Cayman Chemical, Ann Arbor, MI, USA).

## Rhamnolipid measurement

Rhamnolipids in the *P. aeruginosa* EMEM-conditioned medium were measured by adding FM 1-43 (*N*-(3-triethylammoniumpropyl)-4-(4-(dibutylamino) styryl) pyridinium dibromide) (Invitrogen) to a final concentration of 20 µg/mL. FM 1-43 fluorescence was measured on a Cytation5 Multimode Reader (Ex. 475 nm; Em. 595 nm).

## *N*-butanoyl-L-homoserine lactone (C4-HSL) Measurement

To measure C4-HSL concentration, one part spent culture medium from *P. aeruginosa* grown in EMEM was mixed with three parts *E. coli* JM109 pSB536 in the LB medium. This LB medium was inoculated with the *E. coli* reporter strain by diluting an overnight culture fivefold. One hundred fifty microliters of the *P. aeruginosa*-conditioned medium-reporter strain mixture were transferred to each well of a 96-well plate and incubated at 37°C for 2 h in a Cytation5 Multimode Reader. Bioluminescence measurements were taken every 15 min.

## Swarming motility assay

Swarming agar was prepared by supplementing EMEM with 0.5% noble agar. One microliter of *P. aeruginosa* LB overnight culture was dropped onto the agar plate and incubated at 37°C for 18 h. Swarming motility was measured by the area of bacterial growth.

## Pyoverdine purification

An LB overnight culture of *P. aeruginosa* PAO1 was diluted 100-fold into 300 mL of M9 medium [1% wt/vol 5X M9 Salts (BD Difco, Franklin Lakes, NJ, USA), 1.5% wt/vol Bacto Casamino Acids with low iron and salt content (BD Difco), 1 mM $MgSO_4$, and 1 mM $CaCl_2$] in a 2-L flask and grown aerobically for 24 h at 37°C. Bacteria were then removed by centrifugation and filtration through a 0.22 µm membrane. The filtrate was incubated with 10% wt/vol amberlite XAD-4 resin (MilliporeSigma) at room temperature for 4 h with constant agitation. After rinsing the resin with copious amounts of water, pyoverdine was eluted in 50% methanol. This eluent was diluted in water to 15% methanol and loaded onto a Luna Omega 5 µm Polar C18 LC prep column (Phenomenex, Torrance, CA, USA) for high-performance liquid chromatography on a 1220 Infinity LC system (Agilent Technologies, Santa Clara, CA, USA). Pyoverdine was eluted from the column by a 0%–100% methanol gradient across 4 h at a flow rate of 5 mL/min. Fractions were collected every other minute for pyoverdine content analysis (Fig. 5B). The fractions with the highest pyoverdine content were pooled. Methanol was evaporated using a SpeedVac vacuum concentrator. The final purified product was analyzed by HPLC on an analytical column to verify sample purity (Fig. 5C).

## Confocal laser scanning microscopy

Eight million ($8 \times 10^6$) 16HBE cells were seeded into each well of a collagen-coated, six-well plate and were grown at 37°C for ~24 h in a $CO_2$-jacketed incubator until they reached 100% confluence. After treatment, cells were washed in serum-free EMEM and detached from the microtiter plate by trypsin-EDTA solution (MilliporeSigma). After inactivating the trypsin with media containing 10% fetal bovine serum, cells were concentrated via centrifugation and transferred onto a glass side with a 3% noble agar pad. These slides were visualized under an LSM800 AiryScan confocal laser scanning microscope (Zeiss). Pyoverdine fluorescence was visualized via a 405 nm laser line using the channel conditions for Pacific Blue. Dextran-Texas Red (Invitrogen) fluorescence was visualized via a 561 nm laser line using channel conditions for Texas Red. CellMask Deep Red plasma membrane stain (Invitrogen) fluorescence was visualized via a 640 nm laser line using channel conditions for Alex Fluor 660.

## Reverse transcription-quantitative PCR

For *P. aeruginosa*, bacterial cells were collected from 12 mL of EMEM culture by centrifugation. The pellet was resuspended in 2 mL of TRI reagent (Molecular Research Center, Cincinnati, OH, USA) for phenol/chloroform/guanidinium thiocyanate RNA extraction according to manufacturer's protocols (bromochloropropane phase separation followed by isopropanol RNA precipitation). Prior to phase separation, bacterial cells were homogenized with 0.1 mm zirconia beads by vigorous vortexing. To remove DNA contaminants in bacterial RNA extracts, samples were treated with DNase I (Thermo-Fisher Scientific) at 37°C for 30 min, followed by 75°C enzyme heat denaturation for 10 min. For 16HBE cells grown and treated in six-well plates, the treatment medium was aspirated, and cells were incubated in TRI reagent at room temperature for 15 min to lyse cells prior to RNA extraction.

For both bacterial and human cell RNA, cDNA synthesis was performed on a Bio-Rad T100 Thermo Cycler (Bio-Rad, Hercules, CA, USA) using a reverse transcription kit (Applied Biosystems, Waltham, MA, USA). qRT-PCR was performed on a Bio-Rad CFX Connect Real-Time System (Bio-Rad) using a universal qPCR master mix (New England Biolabs, Ipswich, MA, USA). All qPCR primer sequences are shown in Table S1. For *P. aeruginosa* genes, cDNA amplification (Ct value) was normalized to that of housekeeping gene *gyrB*. For 16HBE genes, cDNA amplification was normalized to that of *ACTB*.

## ACKNOWLEDGMENTS

This study was supported by funding from the Cystic Fibrosis Foundation (KIRIEN20I0 to N.V.K.; XU23H0 to Q.X.; KANG19H0 and KANG22H0 to D.K.), the National Institutes of Health (R35GM129294 to N.V.K.), and the American Heart Association (903591 to D.K.).

The authors declare no conflict of interest.

## AUTHOR AFFILIATIONS

[1]Department of BioSciences, Rice University, Houston, Texas, USA
[2]Department of Bioengineering, Rice University, Houston, Texas, USA

## AUTHOR ORCIDs

Donghoon Kang (iD) http://orcid.org/0000-0001-5314-0961
Natalia V. Kirienko (iD) http://orcid.org/0000-0002-1537-4967

## FUNDING

| Funder | Grant(s) | Author(s) |
| --- | --- | --- |
| Cystic Fibrosis Foundation (CFF) | KIRIEN20I0 | Natalia V. Kirienko |
| Cystic Fibrosis Foundation (CFF) | XU23H0 | Qi Xu |
| Cystic Fibrosis Foundation (CFF) | KANG19H0, KANG22H0 | Donghoon Kang |
| HHS | NIH | National Institute of General Medical Sciences (NIGMS) | R35GM129294 | Natalia V. Kirienko |
| American Heart Association (AHA) | 903591 | Donghoon Kang |

## AUTHOR CONTRIBUTIONS

Donghoon Kang, Conceptualization, Funding acquisition, Investigation, Writing – original draft, Writing – review and editing | Qi Xu, Funding acquisition, Investigation, Writing – review and editing | Natalia V. Kirienko, Conceptualization, Funding acquisition, Supervision, Writing – review and editing

## ADDITIONAL FILES

The following material is available online.

### Supplemental Material

**Figures S1-S5 (Spectrum03693-23-S0001.pdf).** Supplemental figures S1-S5 with their legends.
**Figures S6-S10 (Spectrum03693-23-S0002.pdf).** Supplemental figures S6-S10 with their legends.
**Table S1 (Spectrum03693-23-S0003.pdf).** Table S1.

### Open Peer Review

**PEER REVIEW HISTORY (review-history.pdf).** An accounting of the reviewer comments and feedback.

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
