## [Reviewer comments · Microbiology Spectrum]

Microbiology Spectrum

In vitro Lung Epithelial Cell Model Reveals Novel Roles for *Pseudomonas aeruginosa* Siderophores

Donghoon Kang, Qi Xu, and Natalia Kirienko

Corresponding Author(s): Natalia Kirienko, Rice University

Review Timeline:

Submission Date:	October 18, 2023
Editorial Decision:	November 20, 2023
Revision Received:	December 20, 2023
Accepted:	December 21, 2023

Editor: Giordano Rampioni

Reviewer(s): The reviewers have opted to remain anonymous.

Transaction Report:

DOI: <https://doi.org/10.1128/spectrum.03693-23>

Re: Spectrum03693-23 (In vitro Lung Epithelial Cell Model Reveals Novel Roles for Pseudomonas aeruginosa Siderophores)

Dear Dr. Natalia V Kirienko:

Thank you for the privilege of reviewing your work. Your manuscript has been evaluated by two Reviewers with expertise in the area addressed in your study and it was the consensus view of these Reviewers that your manuscript contains interesting data with significant potential impact. However, both Reviewers raised some criticisms that should be addressed before manuscript acceptance. Please also consider that the two-dimensional cellular model used in this study fails to mimic epithelial cell polarity and mucus production, which has a significant role in bacteria / epithelial cell interaction. This limitation should be discussed in the manuscript. Overall, I will be glad to consider for publication in Microbiology Spectrum a revised version of your manuscript addressing the constructive criticisms raised by the Reviewers. Below you will find instructions from the Spectrum editorial office and the reviewer comments.

Revision Guidelines

Sincerely,
Giordano Rampioni
Editor
Microbiology Spectrum

Reviewer #1 (Comments for the Author):

In this study, the cytotoxic effect of pyoverdine and pyochelin was investigated using a human bronchial epithelial cells (16HBE) model by comparing spent media of *P. aeruginosa* wild type and isogenic pyoverdine (pvd) or pyochelin (pch) defective mutants. *P. aeruginosa* pvd mutant supernatant was less cytotoxic than wild type or pch mutant supernatants. In addition, the pyoverdine synthesis inhibitor 5-fluorocytosine reduced the toxicity of high-pyoverdine producers and highly virulent *P. aeruginosa* clinical isolates. However, purified pyoverdine was not cytotoxic.

Since previous studies showed that pyoverdine itself can modulate the expression of other virulence factors, authors postulated that pyoverdine could positively affect the production of other toxins/cytotoxic factors. However, *toxA* or *prpL* genes inactivation did not affect *P. aeruginosa* toxicity. Removal of lipidic fraction from *P. aeruginosa* wild type spent medium by chloroform extraction abrogated cytotoxicity, suggesting involvement of rhamnolipids. In accordance with this hypothesis, purified rhamnolipids were cytotoxic in the 16HBE model.

C4-HSL-dependent QS is a major positive controller of rhamnolipids biosynthesis. However, the pvd defective mutant produced the same C4-HSL levels as the wild type. Authors conclude that pyoverdine modulates the rhamnolipids production independently from QS. Other experiments (less convincing, see major comments) were carried out to rule out the possibility that *mexEF* efflux could play a role in the pyoverdine-dependent regulation of rhamnolipids.

In the final part of the study, authors investigated the effect of different iron chelators on 16HBE inflammatory response. Results showed that pyochelin and other small MW iron chelators (es. deferoxamine) were cytotoxic, Cytotoxicity was abrogated by gallium. However, pyoverdine was not cytotoxic. Interestingly, the combination of pyoverdine and deferoxamine showed increased cytotoxicity compared to deferoxamine alone.

Overall, experiments are well conducted, and results deserves to be published. My major issues are below:

- 1) The work is written in good english but in a convoluted manner. Some logical steps are not made explicit. Each experiment should be described by first explaining why it is done and then what the conclusions are.
- 2) The order in which the experiments are described is not linear and contributes to the reader's confusion. The authors should reorder the description of the results following the order described in the summary above.
- 3) The majority of the experiments concerning the *mexEF* system with transposon mutants are unnecessary and confounding. The only result Authors should describe, in this case, is that *mexT* and *mexEF* genes are not mutated in PAO1 Δ pvdF. The result should be shown in supplementary.
- 4) The secondary metabolite pyocyanin is very cytotoxic and present, together with siderophores and other virulence factors, in membrane vesicles produced by *P. aeruginosa*. Vesicles and their content could be retained by filtering. Authors should consider the possibility that, in addition to rhamnolipids, also pyocyanin production could be reduced in the pvd mutant. Pyocyanin should be measured in *P. aeruginosa* filtered supernatants.
- 5) Levels of *rhIA* transcription should be measured in the wild type and in the pvd mutant.
- 6) Swarming motility is related to virulence and biofilm production, and rhamnolipids play a major role in this phenotype. Authors should compare swarming in wild type and pvd mutant.

Reviewer #2 (Comments for the Author):

Minor comments:

1. A graphical plot of the in vitro model would help to understand much easier the "infection" set up.
2. From line 112 in text , as pvdF-pchBA mutant did not provide further protection compare to pvdF alone, Fig S1 C and E figure should show all statistical comparison including the (NS, if that is the case) between pvdF and pvdF-pchBA double mutant.
3. Line 170 which "mutants" are you describing? Is the PAO1cat vs PAO1pvdF? Please refer it as the control vs the pyoverdine-producing counterparts as in line 157.
4. Line 173 - 174 I am not sure I understand why the authors imply that pyoverdine regulates rhamnolipid production though an alternative pathway when the results were analyzed from simple mutants of *rhII*. The results should be addressed from a double mutant pvdF - *rhII* perspective.
5. Line 189 - think you are referring to Fig S3E and not Fig 3E.
6. WGS raw reads should be deposited into a repositiorium such as SRA from NCBI.
7. A description on how the quantification of lipids was performed, should be included in the material and methods sections.
8. Considering the interesting results, the way by which the authors support their conclusions, and because the role of different molecules is being assessed, I suggest that a graphical diagram showing the most important results achieved in this manuscript is included.

In this manuscript, Donghoon Kang and col., have explored the role of relevant virulence factors from *P. aeruginosa* by using an *in vitro* model of bronchial epithelial cell line monolayer. The project focuses on the host -induced responses and mechanisms involved upon interaction of the host cells with condition media from different mutants of the bacteria. The manuscript includes thorough supporting information for each of the result sections which are supported by several molecular assays such as qRT PCR, pyoverdine production, growth curves, host cell death, as well as confocal microscopy. They also explore the differences between some of the mutants by whole genome sequencing. Data is well integrated and clearly discussed.

Major comments:

My main concern relies on the degree to which the current results can be extrapolated to an *in vivo* scenario, such as the chronic infections that *P. aeruginosa* produce within the lung of patients with cystic fibrosis. A significant number of studies (including this one) have explored the interaction of *P. aeruginosa* and airway epithelial cells using two-dimensional models of cell lines on non-permeable surfaces that do not allow polarization (review Crabbé et al. 2014). Such models, fail to mimic epithelial cell polarity and mucus production, which has a significant advantage for human cells to protect against bacterial invasion. Furthermore, the results and conclusions achieved here could be less drastic or even disappear in the context of a pseudostratified epithelium. The authors should deep-in this matter on the Discussion section.

Minor comments:

1. A graphical plot of the *in vitro* model would help to understand much easier the “infection” set up.
2. From line 112 in text , as pvdF-pchBA mutant did not provide further protection compare to pvdF alone, Fig S1 C and E figure should show all statistical comparison including the (NS, if that is the case) between pvdF and pvdF-pchBA double mutant.
3. Line 170 which “mutants” are you describing? Is the PAO1cat vs PAO1pvdF? Please refer it as the control vs the pyoverdine-producing counterparts as in line 157.
4. Line 173 – 174 I am not sure I understand why the authors imply that pyoverdine regulates rhamnolipid production though an alternative pathway when the results were analyzed from simple mutants of rhII. The results should be addressed from a double mutant pvdF – rhII perspective.
5. Line 189 – think you are referring to Fig S3E and not Fig 3E.
6. WGS raw reads should be deposited into a repositiorium such as SRA from NCBI.
7. A description on how the quantification of lipids was performed, should be included in the material and methods sections.
8. Considering the interesting results, the way by which the authors support their conclusions, and because the role of different molecules is being assessed, I suggest that a graphical diagram showing the most important results achieved in this manuscript is included.

Reviewer #1 (Comments for the Author):

In this study, the cytotoxic effect of pyoverdine and pyochelin was investigated using a human bronchial epithelial cells (16HBE) model by comparing spent media of *P. aeruginosa* wild type and isogenic pyoverdine (pvd) or pyochelin (pch) defective mutants. *P. aeruginosa* pvd mutant supernatant was less cytotoxic than wild type or pch mutant supernatants. In addition, the pyoverdine synthesis inhibitor 5-fluorocytosine reduced the toxicity of high-pyoverdine producers and highly virulent *P. aeruginosa* clinical isolates. However, purified pyoverdine was not cytotoxic.

Since previous studies showed that pyoverdine itself can modulate the expression of other virulence factors, authors postulated that pyoverdine could positively affect the production of other toxins/cytotoxic factors. However, *toxA* or *prpL* genes inactivation did not affect *P. aeruginosa* toxicity. Removal of lipidic fraction from *P. aeruginosa* wild type spent medium by chloroform extraction abrogated cytotoxicity, suggesting involvement of rhamnolipids. In accordance with this hypothesis, purified rhamnolipids were cytotoxic in the 16HBE model. C4-HSL-dependent QS is a major positive controller of rhamnolipids biosynthesis. However, the pvd defective mutant produced the same C4-HSL levels as the wild type. Authors conclude that pyoverdine modulates the rhamnolipids production independently from QS. Other experiments (less convincing, see major comments) were carried out to rule out the possibility that *mexEF* efflux could play a role in the pyoverdine-dependent regulation of rhamnolipids.

In the final part of the study, authors investigated the effect of different iron chelators on 16HBE inflammatory response. Results showed that pyochelin and other small MW iron chelators (es. deferoxamine) were cytotoxic, Cytotoxicity was abrogated by gallium. However, pyoverdine was not cytotoxic. Interestingly, the combination of pyoverdine and deferoxamine showed increased cytotoxicity compared to deferoxamine alone.

Overall, experiments are well conducted, and results deserves to be published. My major issues are below:

1) The work is written in good english but in a convoluted manner. Some logical steps are not made explicit. Each experiment should be described by first explaining why it is done and then what the conclusions are.

We appreciate the reviewer's feedback. We made relevant improvements to the manuscript.

For instance:

- We clearly state the hypothesis before discussing the effects of 5-fluorocytosine: *“Since genetic disruption of pyoverdine biosynthesis decreased toxicity of the conditioned medium, we hypothesized the same result could be accomplished using a chemical inhibitor. To that end, we tested whether the pyoverdine biosynthetic inhibitor and FDA-approved antimycotic drug 5-fluorocytosine (5-FC) inhibited pyoverdine-dependent virulence.”*
- We clearly explain why we are conducting the pyocyanin supplementation experiment: *“We also examined whether conditioned medium cytotoxicity could be attributed to pyocyanin content since pyocyanin is known to cause acute oxidative damage to host cells”*
- We clearly state the hypothesis before discussing the phenotypes of the rhamnolipid biosynthetic mutants in EMEM: *“Based on previous studies, we posited that the relevant secreted lipid factors were rhamnolipids. We recently demonstrated that *P. aeruginosa* secretes*

rhamnolipids that rapidly induce membrane rupture and permeabilization in a wide range of host cells, including murine macrophages, human bronchial epithelial cells, and erythrocytes. To test this hypothesis, we used a rhamnolipid biosynthetic mutant, MPAO1rhlA, to measure the lipid content of conditioned media.”

- We clearly state the purpose of the swarming motility experiment: “*In addition to killing host cells, rhamnolipids are known to regulate P. aeruginosa swarming motility, which promotes pathogen proliferation and biofilm formation within the host. To test whether pyoverdine production affects swarming motility, we measured the area of lawn growth on EMEM semisolid agar (0.5%) for wild-type PAO1, the pyoverdine biosynthetic mutant Δ pvdF, and the rhamnolipid biosynthetic mutant rhlA.*”

2) The order in which the experiments are described is not linear and contributes to the reader's confusion. The authors should reorder the description of the results following the order described in the summary above.

The figures and manuscript text have been rearranged accordingly. As the reviewer suggested, we first discuss the role of pyoverdine in conditioned medium toxicity in **Fig. 1** then discuss how supplementing the *P. aeruginosa* growth medium (EMEM) with 5-fluorocytosine mitigates that toxicity in **Fig. 2**. We then investigate the identity of the toxic factor in the conditioned medium in **Fig. 3** and validate these findings using rhamnolipid biosynthetic mutants in **Fig. 4**. We demonstrate that purified rhamnolipids kill 16HBE cells and that rhamnolipids contribute to swarming motility in EMEM in **Fig. 5**. Finally, we study the consequences of long-term exposure (~24-60 h instead of 30 min as in **Fig. 1-5**) to siderophores (pyoverdine, pyochelin) - their relative toxicities (**Fig. 6**) and their effects on the activation of proinflammatory pathways (**Fig. 7**).

3) The majority of the experiments concerning the mexEF system with transposon mutants are unnecessary and confounding. The only result Authors should describe, in this case, is that mexT and mexEF genes are not mutated in PAO1 Δ pvdF. The result should be shown in supplementary.

We appreciate the reviewer for pointing this out. Based on these concerns, we removed all results regarding the *mexEF* system.

4) The secondary metabolite pyocyanin is very cytotoxic and present, together with siderophores and other virulence factors, in membrane vesicles produced by *P. aeruginosa*. Vesicles and their content could be retained by filtering. Authors should consider the possibility that, in addition to rhamnolipids, also pyocyanin production could be reduced in the pvd mutant. Pyocyanin should be measured in *P. aeruginosa* filtered supernatants.

We appreciate the suggestion. We measured pyocyanin content in the conditioned media (see new materials and methods subsection “*Pyocyanin Measurement*”) and found that WT PAO1 produced significantly greater levels of pyocyanin than the pyoverdine mutant. However, this concentration (~6 μ M) was not sufficient to kill 16HBE cells within the time we observe substantial cell death by the conditioned medium (~15 min). In fact, even concentrations 10-folds greater (60 μ M) was not sufficient to kill cells. These results reinforce our findings that pyoverdine-dependent production of rhamnolipids drives conditioned medium toxicity. We included these results in **Fig. S4** (lines 215-230).

5) Levels of *rhIA* transcription should be measured in the wild type and in the *pvd* mutant.

We measured *rhIA*, *rhIB*, *rhIR*, and *rhII* mRNA levels in WT PAO1 and PAO1 Δ *pvdF* grown in EMEM. For all genes, the pyoverdine mutant did not exhibit substantial (fold change \geq |1.5|) decrease in transcription. We concluded in the manuscript that this may be due to pyoverdine regulating rhamnolipid production via other biosynthetic or regulatory factors or pyoverdine regulating rhamnolipid egress rather than biosynthesis (lines 364-371). The relevant figure has been incorporated as **Fig. S6E**.

6) Swarming motility is related to virulence and biofilm production, and rhamnolipids play a major role in this phenotype. Authors should compare swarming in wild type and *pvd* mutant.

We thank the reviewer for this suggestion. We developed a swarming motility assay for our pathogenesis model by supplementing EMEM with 0.5% noble agar (concentration consistent with numerous other studies). We added a new materials and methods subsection “*Swarming Motility Assay*” describing the procedure. Under these conditions, WT PAO1 exhibited greater swarming behavior than PAO1 Δ *pvdF* or the *rhIA* transposon mutant. The *rhIA* mutant exhibited the least swarming. We included these results in **Fig. 5D, E** (lines 411-418).

Reviewer #2 (Comments for the Author):

Minor comments:

1. A graphical plot of the in vitro model would help to understand much easier the "infection" set up.

We thank the reviewer for the suggestion. We included a graphical diagram of the pathogenesis model in **Fig. 8**. Panels A and C depict the pathogenesis model set up. A: *P. aeruginosa* produces pyoverdine and rhamnolipid-rich conditioned medium when grown in Eagle's Modified Minimum Essential Medium. C: When treated 16HBE cells with this conditioned medium, rhamnolipid interacts with the host membrane, causing acute cell death.

2. From line 112 in text, as *pvdF-pchBA* mutant did not provide further protection compared to *pvdF* alone, Fig S1 C and E figure should show all statistical comparison including the (NS, if that is the case) between *pvdF* and *pvdF-pchBA* double mutant.

We included the additional statistical comparisons in **Fig. S1C, E**. As expected, there was no statistically significant difference between the Δ *pvdF* mutant and Δ *pvdF* Δ *pchBA* double mutant when it came to conditioned medium toxicity towards murine macrophages (**Fig. S1C**) or damage to 16HBE cells (**Fig. S1E**).

3. Line 170 which "mutants" are you describing? Is the PAO1cat vs PAO1pvdF? Please refer it as the control vs the pyoverdine-producing counterparts as in line 157.

We apologize for the confusion. We clarified the strains' identities in lines 358-361:

“Using this reporter, we quantified C4-HSL concentrations in the conditioned medium of pyoverdine mutants (PAO1 Δ pvdF, MPAO1pvdF) and saw no significant difference in C4-HSL production between mutants and their pyoverdine-producing counterparts (WT PAO1, MPAO1cat) (Fig. 4H; Fig. S6A-D).”

4. Line 173 - 174 I am not sure I understand why the authors imply that pyoverdine regulates rhamnolipid production through an alternative pathway when the results were analyzed from simple mutants of *rhII*. The results should be addressed from a double mutant *pvdF* - *rhII* perspective.

We concluded that pyoverdine regulates rhamnolipid production through an alternative pathway because C4-HSL production was unaffected in the two pyoverdine biosynthetic mutants (PAO1 Δ pvdF, MPAO1pvdF – **Fig. S6A-D**). The *rhII* mutant was used to validate the assay. The pyoverdine mutants also did not exhibit substantial downregulation of *rhIR* or *rhII* (**Fig. S6E**).

A double mutant would usually provide insight into the relationship between pyoverdine and C4-HSL production (i.e.: if disruption of pyoverdine biosynthesis further reduced rhamnolipid production in the *rhII* mutant, pyoverdine would be acting independently of *rhII*/C4-HSL). However, this is not applicable in this case because the *rhII* mutation abolishes rhamnolipid production and thus we cannot determine the effects of an additional mutation targeting pyoverdine biosynthesis.

5. Line 189 - think you are referring to Fig S3E and not Fig 3E.

We apologize for the oversight. We corrected the reference in line 409:

*“One caveat of this observation was that disruption of *xcpQ* substantially impaired protease secretion, it was not completely abolished (Fig. S5E).”*

6. WGS raw reads should be deposited into a repository such as SRA from NCBI.

Based on reviewer 1's comments, we removed all results regarding the *mexEF* system including the WGS analysis.

7. A description on how the quantification of lipids was performed, should be included in the material and methods sections.

We included a new materials and methods subsection “*Rhamnolipid Measurement*”.

“Rhamnolipids in the P. aeruginosa EMEM conditioned medium were measured by adding FM 1-43 (N-(3-Triethylammoniumpropyl)-4-(4-(Dibutylamino) Styryl) Pyridinium Dibromide) (Invitrogen) to a final concentration of 20 μ g/mL. FM 1-43 fluorescence was measured on a Cytation5 Multimode Reader (Ex. 475 nm; Em. 595 nm).”

8. Considering the interesting results, the way by which the authors support their conclusions, and because the role of different molecules is being assessed, I suggest that a graphical diagram showing the most important results achieved in this manuscript is included.

We thank the reviewer for this suggestion. We included a graphical diagram in **Fig. 8**.

Re: Spectrum03693-23R1 (In vitro Lung Epithelial Cell Model Reveals Novel Roles for Pseudomonas aeruginosa Siderophores)

Dear Dr. Natalia V Kirienko:

I appreciate your effort to amend the manuscript based on the Reviewers' comments. I am grateful to the Reviewers who did a great job in helping improve this manuscript with their criticisms. I think that the manuscript is now ready for publication. Hence, I am glad to communicate that your manuscript has been accepted, and I am forwarding it to the ASM Journals Department for publication. Your paper will first be checked to make sure all elements meet the technical requirements. ASM staff will contact you if anything needs to be revised before copyediting and production can begin. Otherwise, you will be notified when your proofs are ready to be viewed.

Sincerely,
Giordano Rampioni
Editor
Microbiology Spectrum